# The Tunable Electronic and Optical Properties of Two-Dimensional Bismuth Oxyhalides

**DOI:** 10.3390/nano13202798

**Published:** 2023-10-20

**Authors:** Yong Zhou, Beitong Cheng, Shuai Huang, Xingyong Huang, Ruomei Jiang, Xule Wang, Wei Zhang, Baonan Jia, Pengfei Lu, Hai-Zhi Song

**Affiliations:** 1Quantum Research Center, Southwest Institute of Technical Physics, Chengdu 610041, China; 2School of Electronic Engineering, Chengdu Technological University, Chengdu 611730, China; 3Faculty of Science, Yibin University, Yibin 644007, China; 4State Key Laboratory of Information Photonics and Optical Communications, Beijing University of Posts and Telecommunications, Beijing 100876, China; 5Institute of Fundamental and Frontier Sciences, University of Electronic Science and Technology of China, Chengdu 610054, China; 6State Key Laboratory of High Power Semiconductor Lasers, Changchun University of Science and Technology, Changchun 130013, China

**Keywords:** 2D BiOX materials, density function theory, electronic properties, optical properties

## Abstract

Two-dimensional (2D) bismuth oxyhalides (BiOX) have attracted much attention as potential optoelectronic materials. To explore their application diversity, we herewith systematically investigate the tunable properties of 2D BiOX using first-principles calculations. Their electronic and optical properties can be modulated by changing the number of monolayers, applying strain, and/or varying the halogen composition. The band gap shrinks monotonically and approaches the bulk value, the optical absorption coefficient increases, and the absorption spectrum redshifts as the layer number of 2D BiOX increases. The carrier transport property can be improved by applying tensile strain, and the ability of photocatalytic hydrogen evolution can be obtained by applying compressive strain. General strain engineering will be effective in linearly tuning the band gap of BiOX in a wide strain range. Strain, together with halogen composition variation, can tune the optical absorption spectrum to be on demand in the range from visible to ultraviolet. This suggests that 2D BiOX materials can potentially serve as tunable novel photodetectors, can be used to improve clean energy techniques, and have potential in the field of flexible optoelectronics.

## 1. Introduction

In recent years, research on two-dimensional (2D) materials has attracted significant attention [1,2,3,4,5,6]. These materials—mostly binary compounds, such as transition-metal dichalcogenides [4], metal halides [5], III–V semiconductors [6], etc.—often exhibit a variety of novel physical properties that are distinct from their bulk-phase counterparts [7], holding immense potential for future nano-electronic and optoelectronic applications [8,9,10,11,12,13]. Compared with 2D binary materials, ternary-layered compounds possess more versatility with regard to their physical and chemical properties. Thus, one kind of ternary 2D compound, bismuth oxyhalides (BiOX), has recently drawn increasing attention and has emerged as a noteworthy contender with novel electronic and optical properties [14,15,16,17]. BiOX-related material preparation and device fabrication have also been widely explored. In ref. [18,19,20], 2D BiOX (X = Cl, Br, I) layered materials have been successfully synthesized using the chemical vapor deposition method and microwave technology, yielding high-performance ultraviolet photodetectors, solar cells and potential photocatalysts. However, some drawbacks of BiOX, as found so far—such as relatively large band gaps, weak light absorption in the visible light region, difficulty in matching the water-splitting reaction, etc.—have limited their further application. Effective methods are strongly required to extend the application scope of 2D BiOX.

Two-dimensional materials often exhibit tunable properties capable of widening their utility [21], so the tunability of 2D BiOX is a question worth investigating. Approaches such as heterojunction formation, interface modification, enhancement of Bi content, creation of oxygen vacancies, metal/non-metal doping, layer number control, and strain engineering have provided insights for enhancing the capability of 2D BiOX [22,23,24,25,26,27]. Kong et al. have investigated the electronic structure and optical properties of single-layer BiOI under biaxial strain [28]. These few studies either applied economic simulation methods or were carried out under limited conditions, providing rather insufficient solutions for optimizing the properties of BiOX. On the other hand, compared with 2D binary compounds, more versatile and effective tunability could be reasonably expected. It is thus our motivation to widely and systematically modulate the electronic and optical properties of 2D BiOX.

In this work, we theoretically study the behavior of BiOX by changing the layer thickness, tuning the halogen composition, and engineering the strain. The electronic and optical properties are found to be tunable by changing the number of layers or tuning the proportion of different halogen atoms within BiOX. Strain engineering is an effective way to tune the band gap, band edge, and carrier transport property. It thus sheds light on ways to improve the performance of BiOX 2D materials in the field of optoelectronics.

## 2. Calculation Methods

In this study, all calculations are performed based on the first-principles study with density functional theory (DFT) and the projector augmented wave (PAW) method [29] in the Vienna ab initio simulation package (VASP) [30]. The exchange–correlation function is treated using the Perdew–Burke–Ernzerhof (PBE) form within the generalized gradient approximation (GGA) scheme [31]. The DFT-D3 method [32,33] is adopted to describe the weak van der Waals interaction. A vacuum space of 20 Å is set in the calculation model to avoid interaction with the adjacent layer. The energy cutoff is set as 520 eV, and the convergence criteria of force and energy are set as 0.01 eV/Å and 1 × 10^−5^ eV/atom, respectively. When performing structural optimization, the k-point meshes are set as 9 × 9 × 3 and 9 × 9 × 1 for bulk and layered BiOX, respectively. Since PBE functional usually underestimates band gap [28], the calculations are corrected by the more accurate hybrid function (Heyd–Scuseria–Ernzerhof (HSE06)) [34]. For strain engineering, both uniaxial and biaxial strain are set from −8% (compression) to 8% (tension) with an increment of 2%. The negative (positive) values represent compressive (tensile) strains.

The cleavage energy *E_cl_* of BiOX (X = Cl, Br, I) is calculated as described in [35],
(1)$$E_{cl}=\frac{E_{a}-E_{b}}{S}$$
where *E_b_* and *E_a_* present the energy of systems before and after exfoliation, respectively, and *S* is the cleavage area of the BiOX monolayer. This energy is the minimum required to overcome the interlayer van der Waals coupling in the process of mechanical exfoliation. It can be used to evaluate the feasibility of the experimental preparation.

To estimate the stability of 2D BiOX, phonon spectra calculations are performed on a 5 × 5 × 1 supercell to evaluate their dynamic stability [36,37], and ab initio molecular dynamics (AIMD) simulations are carried out on a 4 × 4 × 1 supercell to investigate thermodynamic stability [21,35]. More specifically, AIMD is simulated under the canonical (NVT) ensemble using the Nosé heat bath scheme, and each simulation lasts for 3000 fs with a time step of 1 fs in reference to similar studies [38,39]. Meanwhile, the cohesive energy *E_c_* is calculated to determine the stability of BiOX by [40],
(2)$$E_{c}=\left(\sum_{i}n_{i}E_{i}\;-\;E_{total}\right)/\sum_{i}n_{i}$$
where *i* means the type of atoms, *n_i_* is the number of type *i* atoms per unit cell, *E_total_* represents the total energy per unit cell, and *E_i_* is the energy of a type *i* atom.

The effective mass in the wave vector *k* direction is described by [41],
(3)$$m*=\frac{\hbar^{2}}{\partial^{2}E/\partial k^{2}}$$
where *ℏ* is the reduced Planck constant and *E* is the total energy.

The optical properties are measured by the frequency $\left(\omega \right)$-dependent complex dielectric function $\varepsilon_{2D}(\omega )=\varepsilon_{1(2D)}(\omega )+i\varepsilon_{2(2D)}(\omega )$, and the refractive spectrum $n_{2D}\left(\omega \right)$ and the optical absorption spectrum $\alpha_{2D}(\omega )$ can be derived from the real part $\left(\varepsilon_{1}\right)$ and the imaginary part $\left(\varepsilon_{2}\right)$ [21,35,37]:(4)$$n_{2D}\left(\omega \right)={\left[\frac{\sqrt{\varepsilon_{1\left(2D\right)}{\left(\omega \right)}^{2}+\varepsilon_{2\left(2D\right)}{\left(\omega \right)}^{2}}+\varepsilon_{1\left(2D\right)}\left(\omega \right)}{2}\right]}^{\frac{1}{2}}$$
(5)$$\alpha_{2D}\left(\omega \right)=\frac{\sqrt{2}\omega }{c}{\left[\sqrt{\varepsilon_{1\left(2D\right)}{\left(\omega \right)}^{2}+\varepsilon_{2\left(2D\right)}{\left(\omega \right)}^{2}}-\varepsilon_{1\left(2D\right)}\left(\omega \right)\right]}^{\frac{1}{2}}$$
where *c* is the speed of light in a vacuum.

## 3. Results and Discussions

### 3.1. Crystal Structure and Stability

BiOX have a unique layered structure [42] in which the [Bi_2_O_2_]^2+^ layer is interleaved by double layers of halogen ions [X]^−^. In the monolayer [X–Bi–O–Bi–X], each Bi atom is coordinated by four oxygen atoms and four halogen atoms, as shown in Figure 1a,b. The optimized lattice constants were calculated as *a* = *b* = 3.907 Å and *c* = 7.492 Å for bulk BiOCl, *a* = *b* = 3.943 Å and *c* = 8.258 Å for bulk BiOBr, and *a* = *b* = 4.02 Å and *c* = 9.271 Å for bulk BiOI, which is in good agreement with previous studies in both experiment and theory [20,22,43]. The calculated lattice constants of BiOX monolayers (3.876, 3.926, and 4.015 Å for BiOCl, BiOBr, and BiOI monolayer, respectively) were also consistent with previous theoretical results [27,43], verifying the validity of the optimization calculations. We found that the optimized lattice constants (*a*) increase slightly with the number of layers, as shown in Table 1. Meanwhile, we designed a new bi-halogen BiOX (X = Cl_0.5_Br_0.5_, Cl_0.5_I_0.5_, Br_0.5_I_0.5_) monolayer structure, as shown in Figure 1c–h, replacing one [X]^−^ layer of BiOX (X = Cl, Br, I) with another halogen atom and maintaining the tetragonal structure. The lattice constants of the BiOX (X = Cl, Cl_0.5_Br_0.5_, Br, Cl_0.5_I_0.5_, Br_0.5_I_0.5_, I) monolayer upon full relaxation by PBE-based DFT calculation, as listed in Appendix A, increased monotonically with the increase in halogen atomic number. It is worth mentioning that, probably owing to the dominant in-plane Bi-O bonding, we did not observe unignorable deformation or bending induced by possible strain [44] related to different upper/lower surfacial halogens, confirming the reasonability of this compositional model.

BiOX possess a layered crystal structure which offers flexibility and the possibility of exfoliation into ultrathin flakes. This flexibility enables the integration of single-layer BiOX into various optoelectronic device architectures. We first tried stressing the cleavage energy *E_cl_* to assess the feasibility of the mechanical exfoliation, as shown in Figure 2, as a function of the distance for a BiOX (X = Cl, Br, I) monolayer to be separated from a five-layered structure. The calculated cleavage energy of monolayers BiOCl, BiOBr and BiOI was 0.455, 0.346 and 0.302 J/m^2^, respectively. With regard to those of graphene (0.37 J/m^2^), single-layer SnP_3_ (0.71 J/m^2^) and δ-InP_3_ monolayer (0.827 J/m^2^) [45,46,47], BiOX monolayer is a fairly typical 2D material with moderate van der Waals interactions, implying that it can be easily prepared via mechanical exfoliation.

There was no imaginary vibration frequency observed in the phonon spectra of the BiOX monolayer, as shown in Appendix A, indicating the dynamic stability of BiOX. Meanwhile, no structure disruption occurred after 3000 fs AIMD simulations at 300 K (Appendix A), demonstrating that BiOX monolayers are thermodynamically stable. In addition, the cohesive energy was calculated as 4.92, 4.77, 4.58, 4.84, 4.74, and 4.67 eV/atom for BiOCl, BiOBr, BiOI, BiOCl_0.5_Br_0.5_, BiOCl_0.5_I_0.5_, and BiOBr_0.5_I_0.5_ monolayers, respectively. This implies that BiOCl is the most stable among the above materials. The cohesive energy of BiOX was greater than that of Bi_2_O_2_S (3.85 eV/atom) [21], p-GeN_2_ (3.96 eV/atom) [48] and Silicene (3.71 eV/atom) [49], indicating their higher stability.

### 3.2. Thickness-Dependent Behavior

Owing to the Van der Waals interlayer interactions, 2D materials usually possess layer-number-dependent electronic properties [50,51]. We investigated the band structure and optical properties of few-layer (two to five layers) BiOX (X = Cl, Br, I) using HSE06 functional [34]. It was first confirmed that the BiOX (X = Cl, Br, I) monolayer exhibits indirect semiconductor band structure, where the CBM is located at the Γ point, while the VBM is located between the Γ and X points in the irreducible Brillouin zone (Appendix A). As shown in Figure 3a, BiOBr was taken as a representor to display the relationship between band structure and thickness (layer number). With increasing layer numbers, the BiOX band structure did not change much, i.e., the CBM always stayed at the Γ point, while the VBM was located along the Γ-X direction. Specifically, the VBM of BiOI moved to and then stayed at the X point from two single layers on. It is thus clear that BiOX maintains an indirect band gap character as the layer number increases, which may be helpful for the material design of BiOX. In the meantime, the calculated band gaps of BiOCl, BiOBr, and BiOI monolayers were 3.745, 3.354, and 2.278 eV, respectively (Appendix A), as is normally expected. With increasing layer numbers, the band gaps did vary, as shown in Figure 3b. They decreased monotonically and gradually approached those of their bulks. The band gaps of BiOCl and BiOBr monolayers were reduced obviously with the layer numbers due to the weak quantum size effect [43,52], but the band gap of BiOI showed less sensitivity to the layer number, which may be due to the relatively large charge transfer and orbital hybridization between Bi and I atoms. According to the projected density of states in Figure 3c, the CBM is dominated by Bi 6p states, while the VBM mainly comprises O 2p and X *n*p (n = 3, 4, 5 for X = Cl, Br, I, respectively) states. Thus, the reduced electronegativity (Cl > Br > I) can explain the decreasing band gaps. As the halogen atom X becomes heavier, the contribution of X *n*p to the density of states becomes more and more obvious, and the interatomic covalent characteristics, from strong to weak, are in this order: Bi–O, Bi–I, Bi–Br, and Bi–Cl. The influence of I on Bi is more obvious than that of Cl and Br, so the deviation of Bi at the CBM in BiOI is more significant than those in BiOCl and BiOBr.

The thickness-dependent optical properties of BiOX (X = Cl, Br, I) was also investigated. At the same layer number, the calculated static dielectric constant *ε*(0) and static refractive index *n*(0) became larger with the increase in atomic number of halogen X (Appendix A), as is normally expected. Figure 4 takes the optical properties of BiOCl as an example to show that the thickness dependence is similar to those of multi-layer Bi_2_O_2_X (X = S, Se, Te) [21] and MoS_2_ [53]. As the layer thickness increased from monolayer to bulk, *ε*(0), *n*(0), and the optical absorption coefficient became larger, the peaks of the imaginary part of the dielectric function and the optical absorption spectrum shifted to the low-energy region (redshift).

Given the above data, which show that the monolayer number is effective in tuning the band gap, and the dielectric constant and light absorption spectrum of 2D BiOX, it may be more achievable to design BiOX-related optoelectronic devices in extended application ranges.

### 3.3. Strain Engineering and Composition Tuning

Strain engineering is often considered an effective technique for tuning the electronic structure of 2D semiconductors [21,54]. Thus, we investigated the tunable properties of 2D BiOX (X = Cl, Br, I) with strain. Figure 5a depicts the band gap variation of BiOX monolayers with uniaxial and biaxial strain, and Appendix A shows the band structures of BiOX monolayers under biaxial strain of −8%, 0, and 8%. Under uniaxial strain, the band gaps of BiOCl and BiOBr monolayers decreased monotonically as the strain varied from −8% compression to 8% tension. In contrast, the band gap of BiOI decreased linearly with the increase in tensile strain; it increased first and then decreased as the compressive strain became larger, with a critical strain at −4%. BiOX monolayers are all indirect band gap semiconductors with or without uniaxial strain, and their CBM always remains at the Γ point. When biaxial strain was applied, the band gaps of BiOCl, BiOBr, and BiOI showed similar variation trends. With the increase in compressive strain, the band gaps increased linearly first and then decreased with a critical strain at −6% and −4% for BiOCl/BiOBr and BiOI, respectively. When the compressive strain increased to −8%, the CBM of BiOX shifted from the Γ point to the M point, and, in the range of compressive strain (−8%, −6%), the CBM and VBM of BiOI were located at the M point and the X point, respectively. With the increase in tensile strain, the band gaps of BiOX decreased monotonically and the CBM of BiOX remained at the Γ point. All of the monotonic changes versus strain appeared almost linear, suggesting high controllability of the 2D BiOX under strain.

In Appendix A, the CBM shows a sharper band curve, implying BiOX (X = Cl, Br, I) monolayers possess smaller electron effective masses. Consistently, the calculated electron (hole) effective masses were 0.28 m_0_ (1.33 m_0_), 0.24 m_0_ (1.12 m_0_), and 0.19 m_0_ (1.96 m_0_) for BiOCl, BiOBr, and BiOI monolayers, respectively. Under tensile strain, the CBM of BiOX monolayer exhibited greater curvature than under compressive strain. It is thus expected that tensile strain will provide a smaller electron effective mass and a better carrier transport property than compressive strain. This is rather helpful to practical performance since tensile strain is more widely used in applications of 2D materials. Under biaxial strain, both BiOBr and BiOI monolayers always exhibited indirect band gaps, regardless of the compressive or tensile strain applied. However, when the tensile strain increased to 8%, we found that the BiOCl monolayer transitioned from indirect band gap to direct band gap, and the CBM and VBM were both located at the Γ point. For both uniaxial and biaxial strain, the band gaps of BiOX monolayers tuned under tensile strain exhibited better linearity, while a critical strain occurred under compressive strain. The critical strain of BiOI is smaller because the smaller electronegativity of I makes it more susceptible to strain. In the strain range of (−4% to 8%), uniaxial and biaxial strain have similar influence on BiOI. BiOCl and BiOBr exhibit better tunability under biaxial strain in the strain range from −6% to 8%. Generally, the band gap of BiOX is found to be tunable in a wide strain range, suggesting application potential in fields such as flexible optoelectronics.

Band edge position is an important factor in applications such as photocatalytic water splitting [35,55,56]. As depicted in Figure 5b, the band edge positions of BiOX (X = Cl, Br, I) monolayers (in reference to the vacuum level) shift downwards under tensile strain, and the downward trend tends to be gentler when the tensile strain becomes larger. With the increase in compressive strain, the VBM position rises while the CBM position shifts first upwards and then downwards with a critical strain at −6%. Adopting appropriate strain is thus a feasible method for adjusting band edge positions. In the range of −2% to −8%, the CBM and VBM positions of strained BiOX shift upwards to higher energy positions with respect to the redox potential levels of water [27], meeting the band edge alignment requirements [56] for photocatalytic water splitting at pH = 0; that is to say, the CBM is higher than the reduction potential of H^+^/H_2_ (−4.44 eV) and the VBM is lower than the oxidation potential of O_2_/H_2_O (−5.67 eV). Obviously, strain engineering has the potential to improve the photocatalytic performance of BiOX at different pH conditions. For instance, water splitting can be switched on/off simply by applying/relaxing a compressive strain, and the efficiency of photocatalytic water splitting at pH = 0 might be controlled by applying compressive strain from −2% to −8%.

The tunability of 2D BiOX was further studied by adjusting the halogen composition *x* in BiOCl*_x_*Br_1−*x*_, BiOCl*_x_*I_1−*x*_, and BiOBr*_x_*I_1−*x*_. Similar to the case *x* = 0.5, as in Figure 1, the models for *x* = 0.75, 0.875 were constructed by replacing one of the 4, 8 halogen atoms with another element in a 2 × 1 × 1, 2 × 2 × 1 supercell, respectively. In unstrained cases, the band gaps decrease gradually as the *x* reduces, making these materials suitable for absorbing a wider range of visible light compared with the original BiOX (X = Cl, Br, I). Combined with strain engineering, tensile strain monotonically reduces the band gap of composition-tuned BiOX monolayers, as shown in Figure 6a–c. When the proportion of I in X is relatively large, the band gap of BiOX is more affected by I, which may be because I is heavier and its bonding with Bi is different from Cl and Br. According to the projected density of states of BiOBr_0.5_I_0.5_ (Figure 6d–f), the unstrained CBM is dominated by Bi 6p orbitals, while its VBM is mainly contributed by I 5p states with a small amount of O 2p states. We found that there are no Br 4p states below the Fermi level, which might be due to the lower energy of Br 4p orbital compared with that of I 5p. With the increase in compressive strain, the contributions of I 5p states to VBM and Bi 6p states to CBM are more prominent. On the contrary, as the tensile strain increases, the contribution of O 2p and Br 4p states to VBM becomes more and more obvious. Therefore, the band gap difference between BiOBr_0.5_I_0.5_ and BiOI becomes greater with the increase in tensile strain. These results indicate that both composition tuning and strain engineering are effective ways to tune the band gap of BiOX (X = Cl, Br, I). It is predicted that composition tuning and/or tensile strain can decrease the band gap and enhance the optical absorption coefficients and the photocatalyst activity of BiOCl and BiOBr in the visible light region. This flexibility in band gap tuning will benefit the material design of 2D BiOX and its applications in optoelectronic devices. On the whole, while those with I are weakly changeable when *x* ≤ 0.5, these BiOX exhibit effective controllability under composition tuning, which is beneficial to many applications.

Next, we tried studying the optical properties of BiOX (X = Cl, Br, I, Cl_0.5_Br_0.5_, Cl_0.5_I_0.5_, Br_0.5_I_0.5_), in light of their importance to optoelectronic materials [18,55]. For unstrained BiOX, the static dielectric constant *ε*(0) and the static refractive index *n*(0) decreased in the order of BiOI, BiOBr_0.5_I_0.5_, BiOCl_0.5_I_0.5_, BiOBr, BiOCl_0.5_Br_0.5_, and BiOCl. This order follows the decreasing average atomic number of halogen X, so it naturally brings about the blueshifting absorption spectrum as shown in Figure 7a. This shift suggests that the absorption spectrum can be tuned on demand in the range from visible to ultraviolet. Specifically, BiOI, BiOBr_0.5_I_0.5_ and BiOCl_0.5_I_0.5_ maintain an absorption coefficient of as high as 10^4^–10^5^ cm^−1^ within the energy range of ~3.1–4.1 eV. These materials may thus be used to compensate the absorption drawback of silicon solar cells in this range to improve the utilization rate of solar energy. Hence, halogen composition tuning enhances the application potential of BiOX in the visible to violet region. With applied strains, the *ε*(0) and *n*(0) change slightly, suggesting a certain stability as the material is tuned by strain. As represented by BiOBr, Figure 7b shows the absorption spectra under strains. It is observed that the optical absorption edges and the absorption peaks redshift under tensile strain, which can improve the ability to absorb visible light. As the compressive strain increases from 0% to −6%, the optical absorption edges blueshift. Together with the fact that the absorption coefficients of strained BiOX in the ultraviolet region are not degraded by compressive or tensile strain, the above properties imply that BiOX has application potential in flexible optoelectronic devices [57].

### 3.4. Discussion

As detailed above, we studied the tunability of 2D BiOX in terms of layer thickness, composition and strain by properly modeling and calculating, and thus predicted their potential applications. Our modeling and calculating methods are not only useful here, but also universal in investigating the tunability of other 2D compound materials, such as transition-metal dichalcogenides, metal halides, MXenes, MOenes, etc., where multiple elements can be incorporated from the same chemical group. We anticipate much more research based on the methods used in this paper.

However, our calculations are presently still somewhat limited. Many aspects, such as the effects of differently strained monolayers on multi-layered BiOX, properties of multi-halogen BiOX, the alloy disorder effect on multi-halogen BiOX, etc., have not been investigated due to the high calculation power required. In addition, DFT calculation with hybrid functional HSE06 should be further optimized as its effect on electron interactions may be distorted in large-scale calculations. These topics are open for further study in the near future.

## 4. Conclusions

In this study, we systematically investigated the properties of 2D BiOX materials. The electronic and optical behaviors of 2D BiOX can be tuned by changing the layer number, varying the halogen composition, and/or applying strain. As the layer number of 2D BiOX increases, the band gap decreases monotonically and gradually approaches that of the bulk material. The optical absorption coefficient also increases, and the absorption spectrum redshifts. Tensile strain can be expected to provide a better carrier transport property, and compressive strain can switch BiOX into materials capable of photocatalyst water-splitting. More generally, strain engineering in a wide range can linearly modulate the band gap of BiOX. Strain application and/or halogen composition variation will be effective to tune the optical absorption spectrum to be on demand in the range from visible to ultraviolet. The observed tunability of 2D BiOX can be expected to extend their application range, to optimize photodetection performance, to improve clean energy technology, and to realize novel flexible optoelectronic devices.

## Figures and Tables

**Figure 1 nanomaterials-13-02798-f001:**
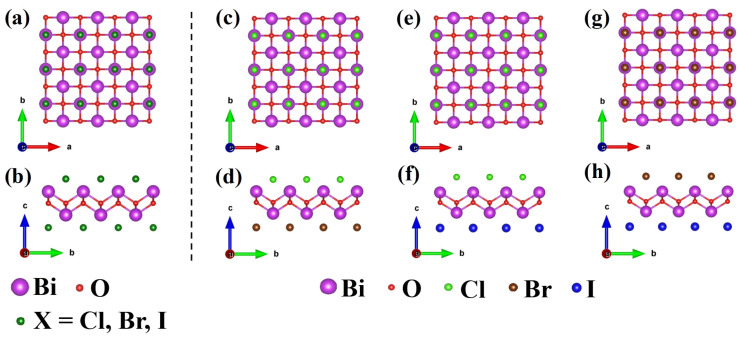
The top view and side view of BiOX monolayer (**a**,**b**) BiOX (X = Cl, Br, I), (**c**,**d**) BiOCl_0.5_Br_0.5_, (**e**,**f**) BiOCl_0.5_I_0.5_, (**g**,**h**) BiOBr_0.5_I_0.5_.

**Figure 2 nanomaterials-13-02798-f002:**
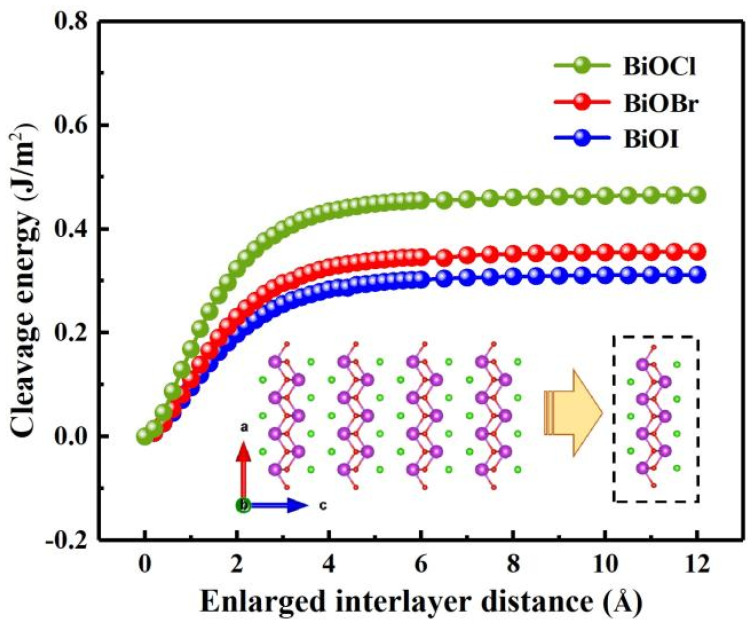
The cleavage energy *E_cl_* (in units of J/m^2^) as a function of the separation distance for five stable BiOX (X = Cl, Br, I) monolayers. Inset: isolating a monolayer from its neighboring four layers.

**Figure 3 nanomaterials-13-02798-f003:**
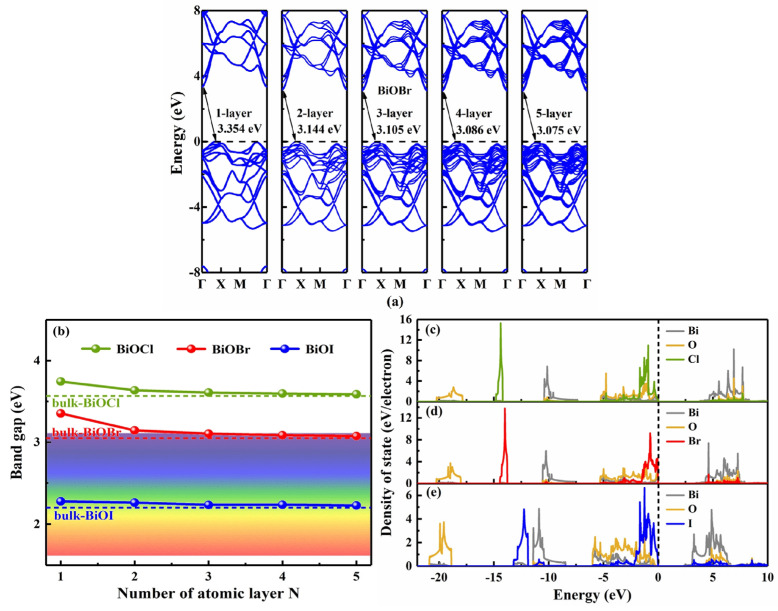
(**a**) The relationship between the HSE06−based band structure of BiOBr and layer numbers (from 1 to 5 layers), (**b**) the variation in band gap with the layer numbers of BiOX (X = Cl, Br, I), and the density of state of (**c**) BiOCl, (**d**) BiOBr, and (**e**) BiOI monolayers.

**Figure 4 nanomaterials-13-02798-f004:**
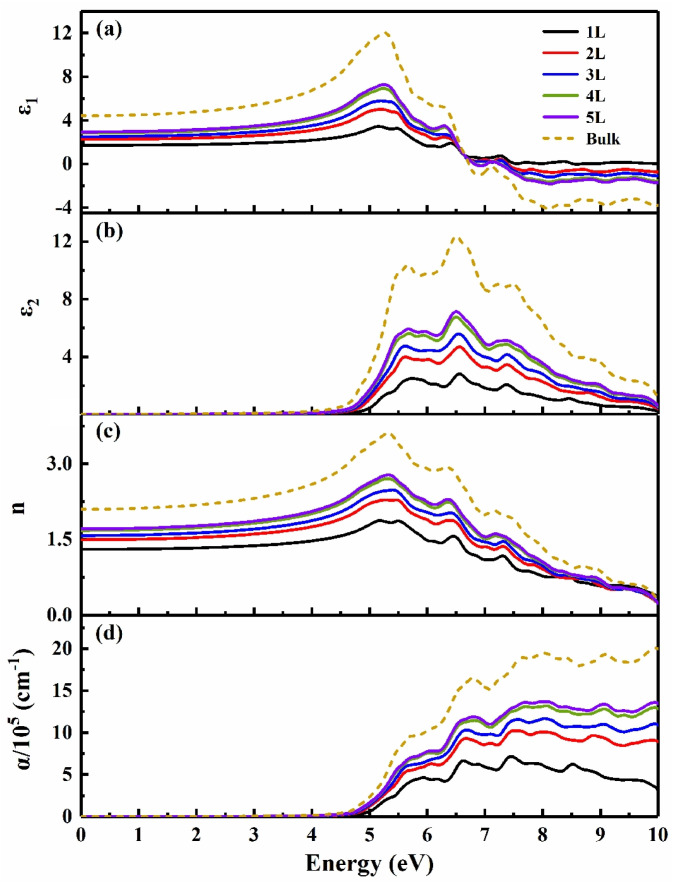
The optical properties of BiOCl (**a**) the real part of dielectric function *ε*_1_, (**b**) the imaginary part of dielectric function *ε*_2_, (**c**) the refractive index *n*, and (**d**) the absorption coefficient *α*.

**Figure 5 nanomaterials-13-02798-f005:**
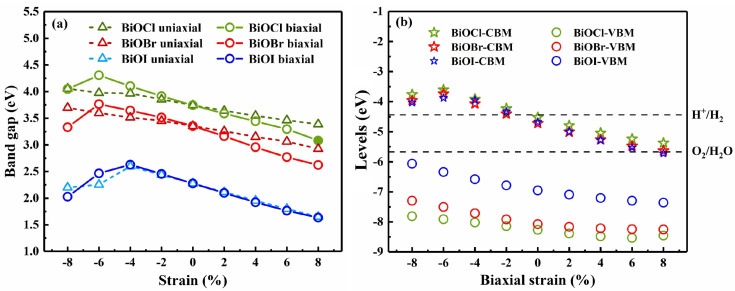
(**a**) The variation in band gap of BiOX (X = Cl, Br, I) with uniaxial and biaxial strain, and (**b**) the relationship between the band edge positions of BiOX (X = Cl, Br, I) and the biaxial strain.

**Figure 6 nanomaterials-13-02798-f006:**
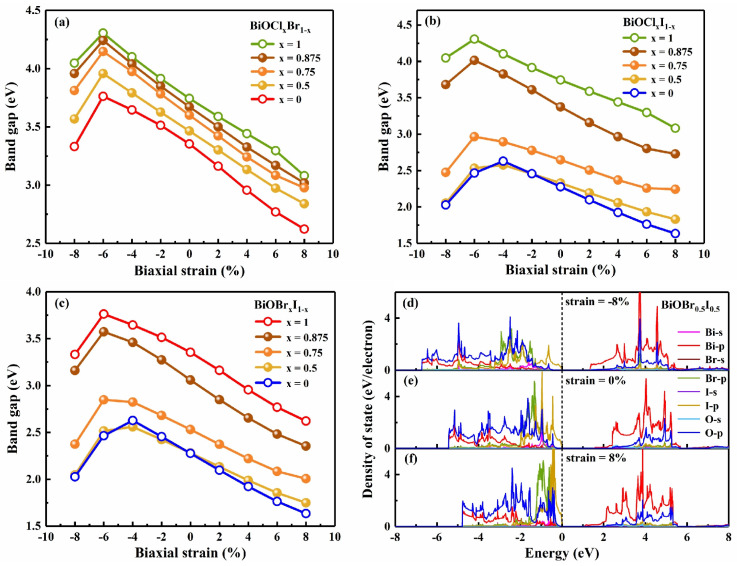
The variation in band gap with the halogen proportion *x* (*x* = 1, 0.875, 0.75, 0.5, 0) and the biaxial strain for (**a**) BiOCl*_x_*Br_1*−x*_, (**b**) BiOCl*_x_*I_1*−x*_, (**c**) BiOBr*_x_*I_1−*x*_, and the density of state of BiOBr_0.5_I_0.5_ under biaxial strain (**d**) −8%, (**e**) 0%, (**f**) 8%.

**Figure 7 nanomaterials-13-02798-f007:**
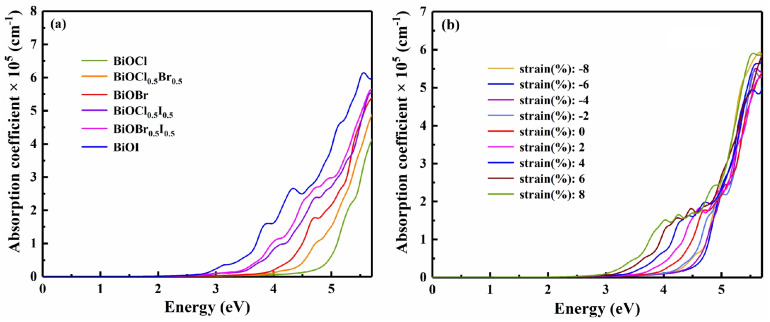
(**a**) The optical absorption spectra of the BiOX (X = Cl, Cl_0.5_Br_0.5_, Br, Cl_0.5_I_0.5_, Br_0.5_I_0.5_, I) monolayer, and (**b**) the variation in the optical absorption spectrum of BiOBr with biaxial strain.

**Table 1 nanomaterials-13-02798-t001:** The calculated lattice parameter constants *a* (Å) of multi-layer BiOX (X = Cl, Br, I).

System	BiOCl	BiOBr	BiOI
	*a* (Å)		
1 Layer	3.876	3.926	4.015
2 Layers	3.891	3.934	4.016
3 Layers	3.896	3.937	4.017
4 Layers	3.899	3.939	4.018
5 Layers	3.900	3.940	4.018
Bulk	3.907	3.943	4.020

## Data Availability

The data presented in this study are available on request from the corresponding author.

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
