# Peer review of "The Tunable Electronic and Optical Properties of Two-Dimensional Bismuth Oxyhalides"

_nanomaterials, 2023, doi:10.3390/nano13202798_

Round 1
Reviewer 1 Report
The work presented in this study involves computational simulations on the behavior of 2D BiOX with varying layer thickness, halogen composition, and strain. The research findings are significant and would be of great interest to scientists working in the field of layered 2D materials. The manuscript is well-written, organized nicely, and the concepts are explained in detail. It meets the requirements of the journal and qualifies strongly for publication in Nanomaterials.
To enhance the quality of this work, it is strongly recommended that the authors include a section on the limitations of the computations they conducted in this study. This section should be added just before the conclusions. Additionally, it would be highly beneficial to include a comment or justification on the universal applicability of this computational study, especially in the area of 2D materials. This will add significant scientific depth to the manuscript.
Author Response
Response to the review comments #1 of manuscript nanomaterials-2630772
October 5, 2023
Dear Editor and Reviewer:
Thank you very much for your letter and for the reviewers’ comments concerning our manuscript entitled “The tunable electronic and optical properties of two-dimensional bismuth oxyhalides” (ID: nanomaterials-2630772). The requirements, evaluations and comments are all very helpful for revising and improving our paper. Accordingly, we rewrote related part of our paper and send you back the revised manuscript with changes hilighted. Here, we would like to respond to the comments and explain how we revise the manuscript. In this letter, the red words represent the changes.
Reviewer #1:
The work presented in this study involves computational simulations on the behavior of 2D BiOX with varying layer thickness, halogen composition, and strain. The research findings are significant and would be of great interest to scientists working in the field of layered 2D materials. The manuscript is well-written, organized nicely, and the concepts are explained in detail. It meets the requirements of the journal and qualifies strongly for publication in Nanomaterials.
To enhance the quality of this work, it is strongly recommended that the authors include a section on the limitations of the computations they conducted in this study. This section should be added just before the conclusions. Additionally, it would be highly beneficial to include a comment or justification on the universal applicability of this computational study, especially in the area of 2D materials. This will add significant scientific depth to the manuscript.
Response to Reviewer #1:
We totally agree on and greatly appreciate the comments.
To address the first comment, we add a paragraph of description on the limitations of our computations, by inserting a sub-section “3.4 Discussions” before the conclusion, and writing the following words into this sub-section. It reads:
“3.4. Discussion
......
However, our calculations are presently somehow limited. Many aspects, such as the effects of differently strained monolayers on multi-layer BiOX, properties of mul-ti-halogen BiOX, alloy disorder effect on multi-halogen BiOX and etc., have not been investigated due to the required high calculation power. In addition, DFT calculation with hybrid functional HSE06 should be further optimized because its result on electrons interaction may be distorted in large scale calculations. These topics are open for our studies in the recent future.”
To responde to the second suggestion, we add a paragraph of description on the universality of our computational study, by inserting the following words before the above paragraph. It reads:
“3.4. Discussion
In the above, by properly modeling and calculating, we studied the tunability of 2D BiOX in terms of layer thickness, composition and strain, and thus suggested their potential applications. Our modeling and computing methods are not only useful here, but are universal in investigating the tunability of many other 2D materials such as transition-metal dichalcogenides, metal halides, MXenes, MOenes and etc., where there can be incorporated multiple elements from one chemical group. We are hoping for much more work based on the methods used in this paper.
......”
We hope these revisions can address what the reviewer strongly concerns.
Thank you very much for your concerning and review arranging.
Sincerely yours,
Hai-Zhi Song
Southwest Institute of Technical Physics
and
University of Electronic Science and Technology of China
Tel: +86-28-68180751(O), +86-15828239155(M)
Email: hzsong1296@163.com, hzsong@uestc.edu.cn
Reviewer 2 Report
Report on “The tunable electronic and optical properties of two-dimensional bismuth oxyhalides”
The authors have studied the electronic and optical properties of two-dimensional bismuth oxyhalides by means of first principles DFT calculations. The subject is interesting and the the simulations have been competently performed using well tested methods.
It is my impression that the manuscript is suitable for publication.
However I have some minor comments which the authors may (or may not) address:
-- In Fig 1 the colours of Br and I seem quite similar to me and perhaps it would be a good idea to change one of them so that the image will be more clear.
-- In the AIMD description although it is mentioned that the total time was set to 3000 fs I didn’t see any reference to the adopted step.
-- I have some concerns with regard to the bi-halogen BiOX (X = Cl0.5Br0.5, Cl0.5I0.5 , Br0.5I0.5). In a BiOX monolayer with the bottom layer of X set to Ionine and the upper layer set to Br, it might be expected that there is going to be an induced strain which in order to be relieved it may lead to a sheet deformation (e.g. bend). However, the use of periodic boundary conditions calculations, would not allow such a deformation to take place. You can see for example App.Phys. Lett. 91, 203112, (2007) where a specific configuration of Si nanowire where shown to be straight using PBC calculations but when PBC were lifted (i.e. finite length NW) the strain was relaxed through bending. If such a behaviour is relevant for bi-halogen BiOX, then the reported values might be affected.
So, perhaps it might be interesting for the authors to check such a hypothesis by performing a geometry optimization on a confined system and check the planarity of the relaxed structure.
The quality of English is generally good.
Author Response
Response to the review comments #2 of manuscript nanomaterials-2630772
October 5, 2023
Dear Editor and Reviewer,
Thank you very much for your letter and for the reviewers’ comments concerning our manuscript entitled “The tunable electronic and optical properties of two-dimensional bismuth oxyhalides” (ID: nanomaterials-2630772). The requirements, evaluations and comments are all very helpful for revising and improving our paper. Accordingly, we rewrote related part of our paper and send you back the revised manuscript with changes tracked. Here, we would like to reply your opinions and explain how we revised the manuscript. In this letter, the red words represent the changes and the blue words represent those omitted.
Reviewer #2:
The authors have studied the electronic and optical properties of two-dimensional bismuth oxyhalides by means of first principles DFT calculations. The subject is interesting and the the simulations have been competently performed using well tested methods.
It is my impression that the manuscript is suitable for publication.
However I have some minor comments which the authors may (or may not) address:
1) In Fig 1 the colours of Br and I seem quite similar to me and perhaps it would be a good idea to change one of them so that the image will be more clear.
2) In the AIMD description although it is mentioned that the total time was set to 3000 fs I didn’t see any reference to the adopted step.
3) I have some concerns with regard to the bi-halogen BiOX (X = Cl0.5Br0.5, Cl0.5I0.5, Br0.5I0.5). In a BiOX monolayer with the bottom layer of X set to Ionine and the upper layer set to Br, it might be expected that there is going to be an induced strain which in order to be relieved it may lead to a sheet deformation (e.g. bend). However, the use of periodic boundary conditions calculations, would not allow such a deformation to take place. You can see for example App.Phys. Lett. 91, 203112, (2007) where a specific configuration of Si nanowire where shown to be straight using PBC calculations but when PBC were lifted (i.e. finite length NW) the strain was relaxed through bending. If such a behaviour is relevant for bi-halogen BiOX, then the reported values might be affected.
So, perhaps it might be interesting for the authors to check such a hypothesis by performing a geometry optimization on a confined system and check the planarity of the relaxed structure.
Response to Reviewer #2:
- We greatly appreciate the comment on the colour distinction between Br and I in Figure 1.
Accordingly, we change the colour of I to blue and redraw Figure 1 in the revised manuscript. We hope it helps to distinguish Br and I more easily.
- We are sorry for that we did not clearly describe the conditions and procedure of the AIMD simulation. In fact, the AIMD simulation was performed on 4×4×1 supercell in the NVT ensemble, lasting for 3,000 fs with a time step of 1 fs. Therefore, we provide a supplementary description about AIMD simulations, add two more references to support the condition setting, and revise related part into the following :
“To estimate the stability of 2D BiOX, phonon spectra calculations are performed on a 5 × 5 × 1 supercell to evaluate their dynamic stability [36,37], and ab initio molecular dynamics (AIMD) simulations are carried out on a 4 × 4 × 1 supercell to investigate the thermodynamic stability [21,35]. In a little more detail, AIMD is simulated under the canonical (NVT) ensemble using Nosé heat bath scheme, and each simulation lasts for 3,000 fs with a time step of 1 fs, refer to similar studies [38,39].”, which appears in lines 86-91 on page 2 in the revised manuscript.
- We appreciate the reviewer’s comment on the possible layer bending. We agree that when the halogens at the upper and lower surfaces of a BiOX monolayer are different, strain imbalance might lead to deformation or bending out of an ideal 2D feature. We also agree that calculations using periodic boundary conditions (PBC) fail to reflect such deformation, like reported (Phys. Lett. 91, 203112, (2007)) or in our work. Accordingly, we try optimizing a 8×1 BiOCl0.5I0.5supercell lattice without PBC condition. The result displayed in Figure 1, however, exhibits no recognizible deformation or bending. This is probably because the in-plane Bi-O bonding in the central layer dominates in the atomic structure formation and there is no obvious in-plane (lateral) interaction related to halogen atoms. As the addressing to this comment, therefore, we add a sentence as “It is worth mentioning that, probably owing to the dominant in-plane Bi-O bonding, we did not observe unignorable deformation or bending induced by possible strain [44] related to different upper/lower surfacial halogens, confirming the reasonability of this compositional model.” to line 119-123 on page 3 in the revised manuscript.
Figure 1. Atomic structure of a chain of 2D BiOCl0.5I0.5 in (a) initial state, (b) optimized state without PBC constrain.
We hope these revisions can satisfy the reviewer’s requirements.
Thank you very much for your concerning and review arranging.
Sincerely yours,
Hai-Zhi Song
Southwest Institute of Technical Physics
and
University of Electronic Science and Technology of China
Tel: +88-28-68180751 (O), +86-15828239155 (M)
Email: hzsong1296@163.com, hzsong@uestc.edu.cn
